# The COVID-19 Pandemic Lowers Active Behavior of Patients with Cardiovascular Diseases, Healthy Peoples and Athletes

**DOI:** 10.3390/ijerph19031108

**Published:** 2022-01-19

**Authors:** Marine Kirsch, Damien Vitiello

**Affiliations:** Institut des Sciences du Sport-Santé de Paris, Université de Paris, F-75015 Paris, France; marine.kirsch@u-paris.fr

**Keywords:** COVID-19, lockdown, cardiovascular disease, detraining, sedentary behavior

## Abstract

Aim: The paper aims to describe the impact of the increasing sedentary lifestyle due to the coronavirus disease-2019 (COVID-19) pandemic restrictions in patients with cardiovascular diseases (CVDs), healthy individuals, and athletes. Methods: A review of studies investigating the impact of the COVID-19 restrictions on patients with CVDs, healthy subjects, and athletes has been conducted in the PubMed, Medline, and Google Scholar medical databases. Results: The review highlighted the significant decrease of active behavior in patients with CVDs and mainly heart-failure patients, illustrated by a reduction of their daily steps and hours of being active during the COVID-19 pandemic. This review also enlightened a significant increase of the time spent in sedentary behavior and the sleep in healthy individuals. Finally, this review reported that the COVID-19 pandemic restrictions induced detraining periods in athletes, altering their health. These periods might also lead to a decrease of their future performances. Conclusions: Staying active and maintaining sufficient levels of physical activity during the COVID-19 pandemic are essential to preserve good health, despite the circumstances of quarantine. Alternatives such as completing a cardiac telerehabilitation for CVD patients or training at home for healthy subjects and athletes may be taken into consideration to maintain a regular active behavior in this sanitary context and potential future pandemics.

## 1. Introduction

Since January 2020, the severe acute respiratory syndrome coronavirus 2 (SARS-CoV-2) has exponentially spread, leading to a worldwide pandemic. The coronavirus disease-2019 (COVID-19) forced the whole world to adapt, in order to contain the outbreak. The governments’ strategies including lockdowns, curfews, working from home, and stay-at-home orders have reduced the possibilities to maintain a regular active behavior.

Since the 1950s, numerous studies have shown that regular exercise is essential to maintain a general good health, but in particular to avoid many pathologies. Indeed, an active behavior is known to have a beneficial effect on cardiovascular health by reducing the overall risk of incident coronary heart disease and stroke [1], to reduce hypertension [2], to increase cardiac output and blood pressure, to lower the resting heart rate and cardiac hypertrophy and to increase myocardial perfusion [3]. Even in the presence of other predictors of cardiovascular mortality, it has been shown in 1996 that death rates were inversely related to cardiorespiratory fitness levels [4].

On the contrary, many factors result in cardiovascular diseases (CVDs), but a sedentary lifestyle with a poor level of exercise is one of the most important. The lack of physical activity leads to an increase of cardiovascular morbidity, arterial stiffness, high density lipoprotein (HDL) cholesterol concentration, mitochondrial dysfunction, and a decrease in lipoprotein lipase activity and induces metabolic disorders [5]. Sedentary behavior is therefore associated with deleterious health outcomes and increased risks of developing a CVD [6].

Given the beneficial effects of being active, physical activity is commonly used in the treatment of CVDs such as chronic heart failure (CHF) and coronary artery disease (CAD). In these conditions, patients may go through a cardiac rehabilitation, which is a customized program of exercise and education, in order to improve patients’ health, to recover from surgery or to treat heart disease. Cardiac rehabilitation is recommended by the American Heart Association, the American College of Cardiology [7] and the European Society of Cardiology [8]. Indeed, research showed its ability to reduce the risk of death from heart disease or future heart problems and to improve the health and quality of life of patients.

Besides the fact that COVID-19 has affected CVD patients, unable to pursue properly their cardiac rehabilitation programs, it has also altered the possibilities of performing physical activity in the general population and athletes through its inherent restrictions.

Therefore, this review aimed at describing the impact of the increasing sedentary lifestyle due to the COVID-19 restrictions in CVD patients, healthy individuals, and athletes.

## 2. Materials and Methods

This systematic review analyzed the impact of the COVID-19 restrictions on patients with CVDs, healthy subjects and athletes. The PubMed, Medline, and Google Scholar databases were examined up to 15 February 2021, using the terms as following: “cardiovascular disease”, “COVID-19”, “detraining” “lockdown”, “quarantine”, and “sedentary behavior”.

The included studies were clinical trials (randomized or not) performed in patients with CVDs and studies performed in healthy subjects and athletes investigating the effects of the COVID-19 lockdown on cardiac rehabilitation participation, active behavior, daily physical activity, detraining, the adherence to the scientific recommendations on physical activity, and sedentary behavior including the time spent being inactive and sleeping.

The study selection is described in Figure 1.

## 3. Results

### 3.1. Impact of the COVID-19 Lockdown in CVD Patients

The COVID-19 outbreak has resulted in a reduction of daily active behavior, due to quarantine restrictions adopted in many countries. Five papers reporting the impact of the pandemic in patients with CVDs have been found (Table 1). In a paper which analyzed the daily number of steps in 26 heart-failure (HF) patients during a six-week period, it has been shown that the step count was significantly lower during each of the first three weeks of the quarantine. The daily steps decreased by over 1100 steps (−16.2%), and among life circumstances, living in a flat or living with at least two other adult persons worsened the deleterious effect of quarantine on the daily step count [9]. In HF patients implanted with cardiac implantable electronic devices (CIEDs), there was a 27.1% decline in the physical activity, and the median physical activity of these patients significantly declined from 2.4 to 1.8 h/day. The change in the trend of the physical activity occurred in the first week of March 2020, which coincides with the beginning of the COVID-19 new measures. The declined amount of the physical activity started to be really significant in the last week of the study, showing that the prolongation of the lockdown has noxious effects on physical activity over time [10].

The COVID-19 pandemic had also an impact on the active behavior in patients with implantable cardioverter defibrillators (ICDs). During the forced 40-day confinement, a mean of a 25% reduction of the physical activity was observed as compared with a 40-day confinement-free period (1.2 ± 0.3 and 1.6 ± 0.5 h/day, respectively) [11].

In an anonymous questionnaire answered by 124 randomly selected CHF patients, it has been reported a 41.9% decrease in the physical activity and a 21.8% increase in HF symptoms such as dyspnea, inferior limb edema, or fatigue. The current pandemic is associated with an increase in unhealthy lifestyle behaviors such as reduced physical activity and reveal deteriorations in well-being and cardiovascular health indexes [12].

In a cohort of 1565 Dutch CVD patients, physical activity and sedentary behavior were compared before and during the COVID-19 lockdown period. The time spent being active declined from 1.0 to 0.0 h/week, the sedentary time increased from 7.8 to 8.9 h/day, and the sedentary behavior increased by 55 min/day [13].

### 3.2. Impact of the COVID-19 Lockdown in Healthy Subjects

Beyond impacts of the COVID-19 restrictions in CVD patients, it also affected the general population’s active behavior. Five papers reporting the impact of the pandemic in healthy subjects have been found (Table 2). It has been reported that the lockdowns resulted from the outbreak have restricted many elements of the environment for university students. During the lockdown, the weekly sitting time increased by 106.76 min [14]. The same results were reported in young adults, as a study showed a significant decline in all physical activity levels such as the vigorous physical activity (before vs. during COVID-19: 9.5 ± 12.5 vs. 6.0 ± 11.6 min/day), the moderate physical activity (before vs. during COVID-19: 11.2 ± 16.0 vs. 5.5 ± 8.7 min/day), and the walking (before vs. during COVID-19: 39.7 ± 30.7 vs. 19.8 ± 24.5 min/day) while an increase in the time spent in both the sedentary behavior (before vs. during COVID-19: 7.8 ± 3.2 vs. 10.0 ± 3.2 h/day) and the sleep (before vs. during COVID-19: 7.7 ± 1.0 vs. 8.4 ± 1.2 h/day) during the COVID-19 outbreak [15].

In a study carried on 3800 healthy adults aged between 18 and 64 years, active behavior decreased significantly during confinement, with a 16.8% reduction of the vigorous activity, a 58.2% reduction of the walking time, and an increase of 23.8% of the sedentary time. The percent of people fulfilling the 75 min/week of the physical activity recommendation decreased by 10.7% [16]. Moreover, a survey of 1047 participants from Asia, Africa, and Europe reported a 33.5% decrease in the number of minutes per day of the physical activity, a decrease in the metabolic equivalents of task (MET) values of 42.7%, and an increase in the sitting time from five to eight hours per day [17]. In addition, 1980 students at six different Bavarian universities took part in a large-scale online survey, and it has been reported that the implementation of the lockdown led to a decrease of the physical activity in 44.5% of the participants. More than 50% and only 39.7% stated to have been exercising for two to five hours weekly before and after the lockdown, respectively. After the implementation of the lockdown, there was a 25% reduction of the daily step count (6777 to 4829 steps per day) [18].

### 3.3. Physical and Physiological Impacts of Training Cessation in Athletes

The COVID-19 pandemic has also obviously affected the world of sports, forcing athletes to train at home. Amateur or professional athletes has then experienced a cessation in their ordinary amounts of physical activity, which has led to a form of detraining. To our knowledge, there is no review aiming at describing the physical and physiological effects of COVID-19 in athletes. Here are summarized the main impacts of detraining reported in athletes that could also appear during lockdowns imposed by the pandemic (Table 3). First, the maximal oxygen uptake (VO_2max_) rapidly declines by between 4% and 14% even after less than four weeks of training cessation [19], by 6% when endurance-trained men stop training for a few weeks [20], and even by up to a 20% reduction in highly trained subjects after three to eight weeks of physical deconditioning [21]. One study reported that three months without formal training sessions reduced the cardiorespiratory fitness of young females athletes to the levels found in non-athletic girls of the same age [22]. Studies have reported that the reduction in cardiovascular function may occur because of a decline in blood volume [20], as an evidence for detraining was a decrease of 5.1 ± 1.9% in the estimated resting plasma volume [23]. Two other studies found a similar 5% decrease [24,25], leading to a decrease of the cardiac preload. However, one study assessed that a 7% decrease of the VO_2max_ was not linked to a decrease in the blood volume [26].

Among studies about detraining, it has been reported an increase of 4% in the maximal heart rate and a 10% decrease of the stroke volume [27]. These changes have negative effects on cardiac output [19,21] and impaired the maximal oxygen delivery capacity. Training cessation have also muscular consequences. Indeed, after only a few days, studies have shown an alteration of mitochondrial function and therefore a decrease of muscle’s oxidative capacity [19,28,29,30,31], inducing detrimental effects on all components of muscular performance such as submaximal strength, the maximal force, or the maximal power in an 8–12 training-cessation period [32]. In addition, eight-week detraining has negative effects on muscle capillarization, as decreases are seen in the number of capillaries per fiber [33]. In a longer detraining period (20 weeks), studies reported a decrease of the mean muscle fiber areas [34] of both fiber types and changes in the distribution and architecture of muscle fibers [35,36]. In power athletes, after 14 days of detraining, the percentage of muscle fiber types and the type I fiber area remain unchanged, but the type II fiber area decreases significantly by 6.4%, suggesting that short-term detraining may affect the size of type II muscle fibers [37]. Moreover, in a study performed on 44 college women trained twice weekly for 10 weeks followed by 5- or 10-week detraining periods, it has been shown that the losses in the maximal values for the oxygen uptake, the oxygen pulse, and the ventilation equivalent were greater for 10 weeks of detraining than for five weeks of detraining [38].

The age of athletes also takes part in detraining adaptations. Indeed, the wall thickness decreased only in young athletes compared to older athletes (6.1 ± 0.5 and 5.7 ± 0.4 mm/m^2^ for young athletes before and after detraining, respectively; 6.3 ± 0.6 and 6.1 ± 0.6 mm/m^2^ for old athletes before and after detraining, respectively). On the contrary, the left-ventricular mass (124 ± 12 and 122 ± 20 g/m^2^ for young athletes before and after detraining, respectively; 134 ± 19 and 113 ± 16 g/m^2^ for old athletes before and after detraining, respectively), end-diastolic diameter (30 ± 3 and 30 ± 1 mm/m^2^ for young athletes before and after detraining, respectively; 29 ± 3 and 28 ± 3 mm/m^2^ for old athletes before and after detraining, respectively) and end-diastolic volume (99 ± 15 and 98 ± 8 mL/m^2^ for young athletes before and after detraining, respectively; 91 ± 17 and 82 ± 13 mL/m^2^ for old athletes before and after detraining, respectively) were only reduced in older athletes [39]. In a study conducted in 70 years old women, a three-month detraining led to negative effects on the resting heart rate (before vs. after detraining: 66.61 ± 6.40 vs. 70.04 ± 6.86 bpm), the systolic blood pressure (before vs. after the detraining: 132.11 ± 9.42 vs. 136.57 ± 10.14 mmHg), the diastolic blood pressure (before vs. after the detraining: 71.86 ± 4.50 vs. 74.82 ± 3.70 mmHg), the pulmonary ventilation (before vs. after the detraining: 44.01 ± 4.42 vs. 39.54 ± 4.93 L/min), and the VO_2_/heart rate (before vs. after the detraining: 13.11 ± 4.93 vs. 11.00 ± 1.90 mL/bpm) [40].

Finally, in a study aiming to quantify the time-magnitude changes in cardiometabolic health outcomes that occur with the cessation of regular exercise training, training adaptations gained in 13 weeks of training were abolished within one month, underlying the importance of a regular physical activity [41]. In addition, after five weeks of detraining, it has been shown a lower adipose tissue and triglyceride delivery during exercise and a worsened metabolic response to exercise with a decreased total fatty acid concentration, inducing a higher glycolysis utilization [42]. Other metabolic changes occur, such as decreased insulin sensitivity and glucose tolerance [43] and increase triglycerides and LDL cholesterol levels [44].

The present summary may help coaches and athletes to better understand the detrimental effects of long periods of training cessation such as the ones imposed by the different lockdowns during the pandemic. This may also help coaches to better monitor precise physical and physiological variables during training cessation periods and to evaluate them at the end of these periods to help athletes to improve return to normal training.

## 4. Discussion

This review reports negative effects of COVID-19 restrictions on active behavior in patients with CVDs and healthy individuals. In CVD patients, this review reported that a reduction of active behavior of patients with ICDs may lead to a loss of metabolic, cardiovascular, and musculoskeletal conditioning in a very little period of time [11] and that the first-wave COVID-19 lockdown increased the sedentary time inducing a net reduction in habitual active behavior levels in Dutch CVD patients [13]. In healthy peoples, this review reported a sedentary lifestyle in young adults during the COVID-19 pandemic [15] associated with a decrease in the daily self-reported physical activity and an increase of the sedentary behavior [16]. Hence, it seems that social isolation and COVID-19 restrictions are likely to lead to a decline of an active behavior in healthy individuals. In athletes, detraining alters physiological adaptations induced by years of training and leads to losses of these adaptations [19]. This suggests that the more the time spent in training cessation, the more the amount of the loss of adaptations occurring [38]. The negative effect of detraining is particularly strong in older athletes [40] and on the maximal oxygen delivery capacity, which may affect endurance capacity as a 4-week detraining period led to a 21% decrease of the endurance capacity in well-trained endurance athletes [45]. Since there is no review on the impacts of COVID-19 on physical and physiological conditions associated with training cessation periods, the summarized effects of detraining reported in this review let us suggest that these negative effects should have affected the physical condition and the physiology of athletes during the different lockdowns imposed by the pandemic.

In CVD patients, maintaining a minimal amount of physical activity is primordial to enhance their fitness. During the pandemic, cardiac prevention and cardiac rehabilitation are essential, as CVD patients are particularly exposed to a worsening of their conditions and COVID-19 can produce or aggravate heart damage [46]. According to different studies, exercise-based cardiac rehabilitation is the most effective preventive program to reduce the burden of cardiovascular risk factors on patients’ health [47,48]. As cardiac rehabilitation’s role is significant to improve the exercise capacity or quality of life in CVD patients [49], it appears to be necessary for patients to be able to continue their programs during the COVID-19 pandemic. The pandemic has led to the closure of many cardiac rehabilitation centers, stopping patients to do so. However, studies plead to a home-based cardiac rehabilitation as an alternative.

Some studies assessed that cardiac telerehabilitation might be a solution to classic cardiac rehabilitation obstacles for participation [50,51], and other studies reported no difference in the effectiveness between classic cardiac rehabilitation and cardiac telerehabilitation [51,52,53]. Home-based cardiac rehabilitation has advantages such as a greater privacy, minimal transportation barriers, and a higher patient independence, low cost, tailoring possible, and protection from virus infection. It also has disadvantages such as less face-to-face interaction, data safety and privacy, lack of legal principles, and lack of social interaction [54]. Despite these possible inconveniences, it has been shown that an elevated participation rate of HF patients in a remote cardiac rehabilitation program during the COVID-19 pandemic reduces the emergency readmission rate. This result suggests that remote cardiac rehabilitation programs can be provided as an alternative to classic cardiac rehabilitation programs [55].

One paper studied the outcomes of peripheral artery disease (PAD) patients enrolled in a structured in-home walking program right before the lockdown to determine whether this intervention ensured the maintenance of mobility even in the case of movement restrictions. It has been reported a best mobility and risk factor control in patients following a structured walking program performed inside home and purposely guided by phone during the COVID-19 pandemic [56]. Thus, home-based cardiac rehabilitation or home-based exercise training programs appear to be interesting alternatives, allowing CVD patients not to interrupt their cardiac rehabilitation programs in order to get all the possible health benefits from them.

In healthy individuals, as the digital world is always evolving, studies indicated that free Internet-based and virtual exercise programs may represent an alternative to in-person exercise options for strength, endurance, and flexibility training [57]. The practice of a regular physical activity is primordial to maintain the quality of life and a general good health status during COVID-19 restrictions, in order to prevent an increasing risk of chronic diseases implied by physical inactivity. Based on the current literature, home-based physical exercise seems to be a good alternative, when restrictions make people unable to exercise outdoor [58].

In athletes, home-confinement appears to lead to detraining and raised a new challenge for sport scientists about how to counteract detraining effects on physiological adaptations relatives to regular exercise and on athletes’ performance. During the different periods of the lockdown, home-based exercise programs have a real development, allowing amateur or professional athletes to maintain a certain amount of physical activity at home. Some papers then brought recommendations and their benefits on health status. For example, increasing the daily step count by only breaking sitting time with two minutes of walking every 20–30 min [59] has a positive impact on mortality and CVD risk [60]. Others recommended to take stairs as much as possible or to do a minimum of 15 min of moderate physical activity per day, as even such little time reduces all-cause mortality [5]. Finally, as physical training should target both the cardiorespiratory system and the skeletal musculature to improve physical fitness and health, the high-intensity interval training (HIIT) appears to be a valuable method to exercise at home. The HIIT is an effective method to improve the maximal oxygen consumption [61], to improve functional capacity and to have beneficial effects on muscle integrity and health status, decreasing the risk to develop a chronic disease [62].

However, to our knowledge, no studies have investigated preventive measures in order to counteract the impact of training cessation in athletes’ performance. Such studies are needed, as future lockdowns can occur in the waiting of an effective COVID-19 vaccine [63] or maybe because of future pandemics.

## 5. Conclusions

The COVID-19 outbreak has led to an important increase of sedentary behavior in patients with CVDs, healthy individuals, and athletes. Staying active and maintaining an active behavior during the COVID-19 pandemic are essential, despite the circumstances of quarantine, and alternatives can be found in training at home or completing a cardiac telerehabilitation for CVD patients. Furthermore, developing sedentary habits can occur rapidly, and concerns turn towards the possibilities of a general maintenance of poor levels of physical activity even after the end of COVID-19 restrictions, resulting in a long-term worsening of people health status. Finding ways to promote the beneficial effects of being regularly active is a challenge for sports scientists, with the aim that the COVID-19 pandemic does not become a worldwide pandemic of physical inactivity and sedentary behavior.

## Figures and Tables

**Figure 1 ijerph-19-01108-f001:**
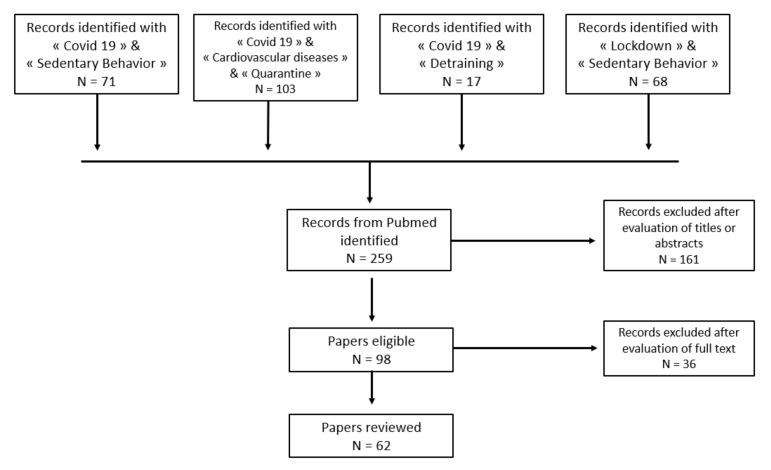
Chart flow of the selection process.

**Table 1 ijerph-19-01108-t001:** Impact of the COVID-19 lockdown in CVD patients.

References	First Author (Year)	Outcome Measures	Results	Conclusion
[9]	Vetrovsky (2020)	The daily numbers of steps in 26 heart failure (HF) patients	A 16.2% decrease of daily steps was found.	Quarantine had a detrimental effect on the level of the habitual physical activity in HF patients.
[10]	Al Faghi (2020)	HF patients with cardiac implantable electronic devices (CIEDs) activity as hours per day from 2 February to 19 April 2020	A 27.1% decline in physical activity was found. The median physical activity significantly declined from 2.4 to 1.8 h/day.	There was a significant decline in the physical activity due to the pandemic.
[11]	Sassone (2020)	The daily physical activities in patients with implantable cardioverter defibrillators (ICDs)	A 25% reduction of the physical activity (1.2 ± 0.3 h/day during the confinement vs. 1.6 ± 0.5 h/day before the confinement)	The COVID-19 pandemic led to an abrupt and statistically significant reduction of the physical activity in patients with primary prevention ICDs.
[12]	Chagué (2020)	The physical activities, the lifestyle behaviors, and the psychological states of 150 randomly selected chronic-heart-failure (CHF) patients	A 41.9% decrease in the physical activity and a 21.8% increase in HF symptoms were found.	The current pandemic had negative effects on lifestyle behaviors such as reduced physical activity.
[13]	Van Bakel (2020)	The physical activity and the sedentary behavior before and during the COVID-19 lockdown period	The time spent exercising declined from 1.0 to 0.0 h/week. The sedentary time increased from 7.8 to 8.9 h/day. The sedentary behavior increased by 55 min/day.	The increase in the sedentary time induced a net reduction in habitual physical activity levels in Dutch cardiovascular diseases (CVDs) patients (48% myocardial infarction) during the first-wave COVID-19 lockdown.

**Table 2 ijerph-19-01108-t002:** Impact of the COVID-19 lockdown in healthy subjects.

References	First Author (Year)	Outcome Measures	Results	Conclusion
[14]	Romero Blanco (2020)	The weekly sitting times in 213 university students	The weekly sitting time increased by 106.76 min.	The confinement changed the physical activities, and the sedentary lifestyles in university students.
[15]	Zheng (2020)	The physical activity levels, the sedentary behaviors, and the sleep in 631 young adults during the COVID-19 epidemic	Walking significantly declined from 39.7 to 19.8 min/day. The times spent in the sedentary behavior and the sleep significantly increased from 7.8 to 10 and 7.7 to 8.4 h/day, respectively.	A significant reduction in physical behaviors and significant increases in the sedentary behavior and the sleep duration of young adults during the COVID-19 epidemic were identified.
[16]	Castaneda-Babarro (2020)	The physical activities, the walking times, and the sedentary times in 3800 healthy adults during confinement	The physical activity and the walking time decreased by 16.8% and 58.2%, respectively. The sedentary time increased by 23.8%.	Healthy adults decreased the daily physical activity and increased the sedentary time during the COVID-19 confinement.
[17]	Ammar (2020)	The physical activities, the lifestyle behaviors, the daily sitting times, and the walking times of 1047 randomly selected adults	The physical activity decreased from five to three days/week. The daily sitting time increased from 5 to 8 h/day. The number of minutes/day walking decreased by 34%.	Home confinement had negative effects on the physical activity with a significant increase in the sitting time, indicative of a more sedentary lifestyle.
[18]	Huber (2020)	The physical activities during the COVID-19 lockdown measures in 1980 students	The physical activity decreased in 44.5% of the participants. The daily step count decreased by 25%.	The COVID-19 crisis led to changes in the physical activity among young adults.

**Table 3 ijerph-19-01108-t003:** Physical and physiological impacts of training cessation in athletes.

References	First Author (Year)	Outcome Measures	Results	Conclusion
[19]	Mujika (2000)	The VO_2max_, the blood volumes, and the maximal cardiac outputs in highly trained athletes after a short-term detraining	Declines in the maximal oxygen uptake (VO_2max_) and the blood volume were found. A reduction of the maximal cardiac output	Short-term detraining induced losses of training-induced physiological and performance adaptations
[20]	Coyle (1986)	The VO_2max_, the cardiac outputs, and the blood volumes in endurance-trained men who stopped training for a few weeks	by 9% in blood volume (5.177 to 4.692 mL), a 12% reduction of the stroke volume, and a 6% reduction of VO_2max_ were found.	The decline in the cardiovascular function following a few weeks of detraining is largely due to a reduction in blood volume.
[21]	Martin (1986)	The oxygen uptakes, the cardiac outputs, the heart rates in 6 exercise-trained endurance athletes after deconditioning	A reduction in stroke volume and a 20% decrease in the left ventricular mass were found.	Inactivity resulted in losses of adaptations such as a greater stroke volume and a regression of left ventricular hypertrophy.
[22]	Raven (1972)	The cardiac outputs and the cardiorespiratory parameters in young females athletes	A reduction of the cardiac output was found.	Three months without formal training sessions reduced the cardiorespiratory fitness of young females athletes.
[23]	Houmard (1992)	The VO_2max_, the resting plasma volumes, and the maximal heart rates in 12 distance runners after 14 days of training cessation	The VO_2max_ decreased by 3 mL/kg/min. The maximal heart rate increased by 9 beats per minute. The resting plasma volume decreased by 5%.	Training cessation affected measures associated with the distance. The running performance was affected by short-term (14 days) training cessation.
[24]	Thompson (1984)	The low-density lipoprotein cholesterol levels of men running 16 km daily after exercise cessation	Low-density lipoprotein cholesterol decreased by 10% to 15%. A 5% decrease in the plasma volume was found.	Exercise cessation led to a reduction in the plasma volume
[25]	Cullinane (1986)	The maximum oxygen uptakes, the estimated changes in the plasma volume, and the cardiac dimensions of 15 male competitive distance runners before and after 10 days of exercise cessation	The plasma volume decreased by 5%. The resting heart rate, blood pressure, and cardiac dimensions remained unchanged with the physical inactivity.	Short periods of the exercise cessation decrease estimated the plasma volume and increased the maximum exercise heart rate of endurance athletes but did not alter their cardiac dimensions.
[26]	Raven (1998)	The VO_2max_ and the lower body negative pressures in 19 volunteers after an 8-week physical deconditioning	The VO_2max_ and the lower body negative pressure tolerance decreased by 7% and 13%, respectively.	The functional modification of the cardiac pressure–volume relationship resulted in the reduced lower body negative pressure tolerance.
[27]	Coyle (1984)	The maximal heart rates, the stroke volumes, and the VO_2max_ in 7 endurance exercise-trained subjects after the cessation of training	VO_2max_ declined by 7% during the first 21 day of inactivity. An increase of 4% in the maximal heart rate was found. A decrease of 10% of the stroke volume was identified.	Loss of adaptations after stopping prolonged intense endurance training occurred from 21 days.
[28]	Coyle (1985)	The heart rates, the ventilations, the respiratory exchange ratios, and the blood lactate concentrations in 7 endurance-exercise-trained subjects after the cessation of training	After 84 days of detraining, experimental subjects’ muscle mitochondrial enzyme levels were still 50% above, and the lactate dehydrogenase (LDH) activity was 22% below sedentary control levels.	Adaptations to prolonged endurance training (responsible for the higher lactate threshold) persisting for a long time after training were stopped.
[29]	Wibom (1992)	The mitochondrial ATP production rates in 9 men after 3 weeks of detraining	The mitochondrial ATP production rate decreased by 12–28%.	Mitochondrial ATP production rate decreased with detraining.
[30]	Henriksson (1977)	Succinate dehydrogenase (SDH) and cytochrome oxidase activities during a 6-week period without training	SDH and cytochrome activities returned to the pre-training level.	The fast return to the pre-training levels of both SDH and cytochrome oxidase activities indicated a high turnover rate of enzymes in the TCA cycle as well as the respiratory chain.
[31]	Moore (1987)	The VO_2max_ and the citrate synthase (CS) activities in trained subjects after 3 weeks of inactivity	A decrease in CS activity to 80 ± 14.6 nmol/mg protein/min was found.	The mitochondrial content of working skeletal muscle is an important determinant of the substrate utilization during submaximal exercise.
[32]	Bosquet (2013)	Meta analysis to assess the effect of resistance training cessation on the strength performance	The submaximal strength, the maximal force, and the maximal power declined.	Resistance training cessation had detrimental effects on all components of muscular performance.
[33]	Klausen (1981)	The numbers of capillaries per mm^2^ and the numbers of capillaries per fiber in 6 male subjects after 8 weeks of detraining	The number of capillaries per fiber decreased.	Eight-week detraining had negative effects on muscle capillarization.
[34]	Psilander (1985)	The myonuclear numbers, the fiber volumes, and the cross-sectional areas (CSAs) assessed in 19 subjects after 20 weeks of detraining	The CSA decreased to 17%.	Long detraining periods led to a decrease of the mean muscle fiber areas.
[35]	Häkkinen (1981)	The maximal isometric strengths, the strengths correlated, and neural activations in 11 males after 12 weeks of detraining	A decrease of the maximal isometric strength and a decrease of the mean muscle-fiber areas of both fiber types were identified.	Detraining affected muscle hypertrophy.
[36]	Houston (1979)	Activities of SDH and LDH, the VO_2max_, and the muscle fiber areas in 6 well-trained runners after 15 days of detraining	SDH and LDH activities decreased by 24% and 13%, respectively. The VO_2max_ decreased by 4%. The muscle fiber areas became larger.	Short periods of detraining resulted in significant changes in indices of physiological capacity and function.
[37]	Hortobagyi (1993)	The performances, the surface EMG activities, and the types of fibers in 12 power athletes after 14 days of detraining	The performances declined. Type II fiber area decreased by 6.4%.	Short-term detraining affected the size of the type II muscle fibers.
[38]	Fringer (1974)	The pulmonary ventilations, the oxygen uptakes, the oxygen pulses, the heart rates, and the total work outputs in 44 trained women after 5 or 10 weeks of detraining	Increases in the resting heart rate and the maximal ventilation equivalent were found. Decreases in the total work, the pulmonary ventilation, the oxygen uptake, and the oxygen pulse were identified.	Losses in the maximal values for the oxygen uptake, the oxygen pulse, and the ventilation equivalent were greater for 10 weeks of detraining than for 5 weeks of detraining.
[39]	Giada (1998)	The left ventricle morphologies, systolic functions, and diastolic filling patterns of 24 male cyclists, 12 young, and 12 older, after a 2-month detraining	The wall thicknesses decreased only in young athletes, while the left ventricular mass and the end-diastolic diameter and volume reduced only in older athletes.	Detraining induced greater left ventricular morphological modifications in older athletes.
[40]	Leitão (2019)	The oxygen uptake (VO_2_) and health profile assessments in 47 older trained women after 3 months of detraining	Increases of the resting heart rate and the systolic and diastolic blood pressures were found. Decreases of the pulmonary ventilation and the VO_2_/heart rate were identified.	Detraining induced greater declines in the total health profile and in VO_2_ after a training particularly developed for older women.
[41]	Nolan (2018)	The VO_2max_, the body fat percentage, the mean arterial pressure, and the HDL cholesterol and triglycerides levels after a 13-week training program followed by detraining	The VO_2max_ and the body fat percentage, along with the mean arterial pressure and HDL cholesterol and triglycerides levels, significantly worsened.	These novel findings underscored the importance of sustained and uninterrupted exercise training.
[42]	Petitbois (2003)	The VO_2max_ and the metabolic responses in 10 trained rowers after detraining	A lower adipose tissue triglyceride delivery during exercise was found. The total fatty acid concentration decreased.	Alterations of the metabolic adaptations to training may become rapidly chronic after such a detraining.
[43]	Heath (1983)	The VO_2max_ values, the glucose tolerances, and the insulin sensitivities in 8 well-trained subjects who stopped training for 10 days	The maximum rise in the plasma insulin concentration was 100%. Blood glucose concentrations higher	Detraining induced decreased the insulin sensitivity and the glucose tolerance
[44]	Giada (1995)	The VO_2max_, the total, LDL, and HDL cholesterol level, and the triglycerides levels in 24 males cyclists after a 2-month detraining	The VO_2max_ decreased. The triglycerides and LDL cholesterol levels increased.	Detraining induced changes in metabolic response to exercise.

## Data Availability

No relevant for this manuscript.

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
