# Peer review of "The COVID-19 Pandemic Lowers Active Behavior of Patients with Cardiovascular Diseases, Healthy Peoples and Athletes"

_ijerph, 2022, doi:10.3390/ijerph19031108_

Round 1
Reviewer 1 Report
The study presented highlights the negative influence that COVID-19 has had on athletes, on the healthy population in general and on patients with cardiovascular diseases. In this way, it focuses on the importance of promoting an active life among citizens. A priori, the study is interesting, but it is necessary to prove that an adequate methodology has been followed for the selection of the reviewed articles and for their analysis.
In the Material and Methods section it is not specified whether they have carried out a systematic review. To accredit a quality review, it would be necessary to follow some validated guide such as The Prisma 2020 Statement (Page, M.J.; McKenzie, J.E., Bossuyt, P.M.; Boutron, I.; Hoffmann, T.C.; Mulrow, C.D.; Shamseer, L.; Tetzlaff, J.M.; Akl, E.A.; Brennan, S.E.; et al. The PRISMA 2020 statement: an updated guideline for reporting systematic reviews. BMJ. 2021, 29, 372:n71. doi: 10.1136/bmj.n71. PMID: 33782057; PMCID: PMC8005924).
The 62 reviewed papers are not presented in the results section. Only 34 publications are counted. Authors must present all articles so that the reader, if deemed necessary, can analyze them in more detail.
The results section is divided into three parts: Impact of Covid-19 lockdown in CVD patients, Impact of Covid-19 lockdown in healthy subjects, and Impact of Covid-19 lockdown in athletes. In section 3.1 a table is included that allows viewing the most relevant information of the 5 articles found, however, in sections 3.2 and 3.3 the explanatory table is not included.
In the list of references there are only 62 publications, which would be insufficient, if they correspond to those included in the review, since at least a different bibliography should be included in the discussion to discuss the results found.
The references are not ordered according to their appearance in the text. From reference 13 it jumps to 40. From reference 44 it jumps to 14. It is possible that in the final writing of the manuscript they have reversed the order of sections 3.2 and 3.3. On the other hand, reference 23 appears for the first time in the discussion, when by number it should be in section 3.3. The authors must adjust the order of these references.
The conclusions are presented in a clear manner but it is necessary to remedy what has been said above with regard to the results and discussion section to confirm that the conclusions are well founded.
A section on the limitations of the study should be included, as well as its practical applications, which is appreciated by the journal and its potential readers. The article is adequately and clearly written. There are no self-citations by the authors.
They should review the journal articles references to adjust them to the style recommended by the journal. According to the Journal instructions for authors, references should be described as follows, depending on the type of work:
- Journal Articles:
- Author 1, A.B.; Author 2, C.D. Title of the article. Abbreviated Journal NameYear, Volume, page range.
Author Response
Thank you for your review of our article.
Please see the attachment file for our responses.

Reviewer 2 Report
The authors are examining changes in physical activity and sedentary behavior associated with COVID-19 related restrictions. Specifically, the authors are reviewing literature that has reported on the impact of COVID-19 restrictions on the physical activity among patients with CVD and among healthy adults.
Major Comments:
The authors could include more details about the assessments of physical activity and sedentary time that were used in the articles reviewed. Specifically they could provide more details about both the method of evaluation (self-report survey, steps as recorded on smart phone, etc) and the nature of the pre-post assessment. Under what circumstances was the pre assessment made (retrospective self-report, other). Understanding the latter is essential in understanding the limits to generalizability of the findings.
The authors need to clarify if the literature related to athletes was focused on data examining conditioning before and after COVID-19 restrictions. A general review of the literature on deconditioning of athletes does not appear aligned with the proposed manuscript and would belong in a separate manuscript. If the articles reviewed explain the impact of COVID-19 restrictions on athlete’s conditioning greater details are needed to help the reader understand pre-post assessments as well as the nature of the restrictions/limitations to the training.
Based on the author’s review of the literature do we need better ongoing physical activity surveillance in order to better assess changes in physical activity related to public health challenges? Does this vary by country?
The discussion introduces a substantial amount of new information regarding the use of remote programs and telewellness. It seems this information deserves a section in the results. The discussion can then focus on synthesizing information previously presented.
Page 6, It is not clear how “digital literacy” is a disadvantage of home-based cardiac rehabilitation. Home-based tellewellness needs to ensure it can engage those with low digital literacy. Consider revisiting the sentence starting “It also has disadvantages…”
Page 7 A reference is needed for the sentence starting “Based on the current literature, home-based physical exercise…” .
Consider standardizing terms used in the article: physical activity, physical exercise, active behavior, etc. Then ensure the title is aligned with this standardization.
COVID-19 as a reference to the current pandemic should be in all capital letters.
Minor Comments:
Page 3, remove “At last…” and start the paragraph “In a cohort…”
Page 6 “Some studies already assessed” . The time reference is not clear, consider removing “already”.
Page 6 consider “other studies reported no difference in the effectiveness between classic…”
Page 6 consider “can be provided as an alternative to the …”
Page 7 consider “studies indicate that free internet-based and virtual exercise programs may represent an alternative to in-person exercise options for strength, endurance, and flexibility training”
Author Response

(The authors gave the same response as above.)

Round 2
Reviewer 2 Report
The heading “3.3. Impact of COVID-19 lockdown in athletes” does seem aligned with the manuscript. That heading suggests the authors will review data regarding changes that specifically occurred among athletes during COVID-19 lockdown. Sections 3.1 and 3.2 appear to report on data collected during COVID-19 lockdown. It appears that the papers reviewed in 3.3 were published pre-pandemic. Better alignment/clarification is needed regarding section 3.3.
